# Function, Mechanism, and Application of Plant Melatonin: An Update with a Focus on the Cereal Crop, Barley (*Hordeum vulgare* L.)

**DOI:** 10.3390/antiox11040634

**Published:** 2022-03-25

**Authors:** Xinxing Yang, Jie Chen, Yuan Ma, Minhua Huang, Ting Qiu, Hongwu Bian, Ning Han, Junhui Wang

**Affiliations:** 1Institute of Genetics and Regenerative Biology, College of Life Sciences, Zhejiang University, Hangzhou 310058, China; xinxingyang@zju.edu.cn (X.Y.); 21907036@zju.edu.cn (J.C.); 22107022@zju.edu.cn (Y.M.); 22007031@zju.edu.cn (M.H.); hwbian@zju.edu.cn (H.B.); ninghan@zju.edu.cn (N.H.); 2School of Pharmacy, Hangzhou Normal University, Hangzhou 311121, China; tingqiu@hznu.edu.cn

**Keywords:** antioxidant, melatonin biosynthesis, barley (*Hordeum vulgare* L.), plant stress response, CAND2/PMTR1

## Abstract

Melatonin is a multiple-function molecule that was first identified in animals and later in plants. Plant melatonin regulates versatile processes involved in plant growth and development, including seed germination, root architecture, flowering time, leaf senescence, fruit ripening, and biomass production. Published reviews on plant melatonin have been focused on two model plants: (1) Arabidopsis and (2) rice, in which the natural melatonin contents are quite low. Efforts to integrate the function and the mechanism of plant melatonin and to determine how plant melatonin benefits human health are also lacking. Barley is a unique cereal crop used for food, feed, and malt. In this study, a bioinformatics analysis to identify the genes required for barley melatonin biosynthesis was first performed, after which the effects of exogenous melatonin on barley growth and development were reviewed. Three integrated mechanisms of melatonin on plant cells were found: (1) serving as an antioxidant, (2) modulating plant hormone crosstalk, and (3) signaling through a putative plant melatonin receptor. Reliable approaches for characterizing the function of barley melatonin biosynthetic genes and to modulate the melatonin contents in barley grains are discussed. The present paper should be helpful for the improvement of barley production under hostile environments and for the reduction of pesticide and fungicide usage in barley cultivation. This study is also beneficial for the enhancement of the nutritional values and healthcare functions of barley in the food industry.

## 1. Introduction

Melatonin, a small chemical that is capable of reversing the darkening effects of melanocyte-stimulating hormones (MSHs) in vertebrates, was first isolated from the bovine pineal gland and given the name by Aaron Lerner and his coworkers in 1958 [1]. Subsequently, the precise structure of melatonin (*N*-acetyl-5-methoxytryptamine) and its biosynthesis pathway from tryptophan metabolism was characterized [2,3,4,5]. From then on, a great number of studies on its physiological roles and molecular mechanisms playing on vertebrates, especially on mammals, were conducted. As a general conclusion, this molecule serves both as a neurohormone and a free radical modulator in numerous cellular processes during the regulation of body temperature, blood pressure, mood, the immune system, and circadian rhythm [5,6,7,8].

Melatonin modulates the sleep and wakefulness cycle in mammals by activating two high-affinity G-protein-coupled melatonin receptors: (1) MT_1_ and (2) MT_2_. Recently, the crystal structures of the human MT_1_ and MT_2_ receptors in complexes with certain melatonin agonists and analogs were revealed [9,10]. These findings could facilitate the design of future compounds in addition to therapeutic agents for the treatment of sleep and mood-related disorders [9,10]. Indeed, in a more recent study, researchers docked more than 150 million virtual molecules to the MT_1_ crystal structure and found 38 high-ranking molecules with potential activation abilities to modulate the biology of melatonin receptors [11]. Contrary to MT_1_ and MT_2_, the third melatonin binding site, MT_3_, is currently under debate as the biochemical nature of MT_3_ appears to be as same as that of quinone reductase 2 [12].

For a long time, melatonin was thought only to be present and execute functions in vertebrates. In 1995, three independent teams reported that melatonin is naturally distributed in plants [13,14,15]. Henceforth, research studies examining plant melatonin have expanded into an exciting field for plant biology. A set of recent review articles have expatiated the biosynthesis, distribution, function, and mechanism of plant melatonin [16,17,18,19,20,21,22]. It is generally acknowledged that melatonin is a multiple-function molecule that is capable of regulating versatile pathways in plant growth and development, including seed germination, root architecture, flowering time, leaf senescence, fruit ripening, and biomass production, together with biotic and abiotic stress tolerance. Nevertheless, these reviews were basically focused on two model plants: (1) Arabidopsis (*Arabidopsis thaliana*) and (2) rice (*Oryza sativa*), in which the natural melatonin contents are quite low. Efforts to integrate the function and mechanism of plant melatonin and to examine how plant melatonin benefits human health are also required.

Barley (*Hordeum vulgare*) is a crop that ranks fourth in total cereal production, just after wheat (*Triticum aestivum*), rice, and maize (*Zea mays*) [23,24]. Many reasons for the widespread cultivation of barley exist. Barley has excellent adaptability to natural environments. Barley can tolerate a wide range of growth conditions, for instance, the high altitudes of the Himalayan foothills. It also has a unique place in the beer manufacturing industry. In addition, with the abundant accumulation of certain special nutrients with healthcare functions, such as β-glucan, vitamin E, and γ-amino butyric acid, barley is an ideal cereal for food and animal feed. In recent years, because of its many health benefits, including a reduction in the risk of heart disease, lowering of blood cholesterol levels, increasing insulin responses in diabetics, and preventing obesity and cancer, a preference for barley has been acquired by a growing number of people [23,24].

In this study, bioinformatics analysis to identify the genes required for the barley melatonin biosynthesis pathway was first conducted. A review of the effects of exogenous melatonin on barley growth and development was then performed. In addition, three integrated mechanisms of melatonin on plant cells are described: (1) serving as an antioxidant, (2) modulating plant hormone crosstalk, and (3) signaling through a putative plant melatonin receptor. Finally, reliable approaches for characterizing the function of barley melatonin biosynthetic genes and modulating the melatonin contents in barley grains are discussed. The present paper should be helpful in several research areas: (1) ways to increase barley production under hostile environments, (2) ways to reduce the usage of pesticides and fungicides in barley cultivation, and (3) ways to enhance the nutritional value and the healthcare function of barley for better use in food, feed, and malt.

## 2. Biosynthesis of Melatonin in Barley

The melatonin biosynthesis pathways of animals and plants share some similarities [5,20,21,25,26]. In both cases, melatonin’s precursor is tryptophan, and the two pathways involve a common serotonin intermediate (Figure 1). Two major differences between the animal and the plant pathways have been found. First, the two enzymes that operate successively to catalyze the conversion of tryptophan into serotonin are different: (1) animal cells adopt tryptophan hydroxylase (TPH) and aromatic amino acid decarboxylase (AADC) and (2) plant cells employ tryptophan decarboxylase and tryptamine 5-hydroxylase (TDC and T5H, respectively), as shown in Figure 1. Thus, the order of the hydroxylation and decarboxylation steps is opposite in animals and plants. Second, to transform serotonin into melatonin, the acetylation and methylation steps also present some differences in plants and animals. Animal cells successively use the animal-type serotonin *N*-acetyltransferase and *N*-acetyl serotonin methyltransferase (SNAT and ASMT, respectively), as shown in Figure 1. Notably, mammalian SNAT and ASMT are also known as arylalkylamine *N*-acetyltransferase and hydroxylindole-*O*-methyltransferase (AANAT and HIOMT, respectively). Plant cells have dual orders for the conversion of serotonin: (1) acetylation followed by methylation or (2) methylation followed by acetylation [21,26,27] (Figure 1). In addition, plants usually use caffeic acid *O*-methyltransferase (COMT) to fulfill the function of ASMT. Furthermore, plant cells have an enzyme termed *N*-acetylserotonin deacetylase (ASDAC) to catalyze the reverse reaction of serotonin acetylation [21,26,27] (Figure 1). As an important finding, the crystal structural basis of plant SNATs during catalysis has recently been elucidated [27]. Plant SNATs form dimers in solution, a feature totally different from the animal SNAT/AANAT [27]. This finding explains the fact that no sequence conservation between the plant and animal SNATs exists.

In plants, chloroplasts are the sole organelles for tryptophan biosynthesis and the major sites for melatonin biosynthesis [21,26], while, in mammals, tryptophan is one of the essential amino acids obtained from dietary sources, and melatonin is produced both in the cytoplasm and mitochondria [25].

In the Kyoto Encyclopedia of Genes and Genomes (KEGG) database resource (https://www.kegg.jp, accessed on 20 December 2021), rice is used as a model plant to illustrate the plant melatonin biosynthesis pathway (module M00936). Proteins orthologous to rice TDC, T5H, ASMT, and SNAT from more than 50 plant species have been included in the database, but no information about barley has been published. On the other hand, in a plant metabolic cycle database termed the Plant Metabolic Network (PMN), the barley genes putatively involved in the melatonin biosynthetic pathway have been computationally predicted (https://plantcyc.org, version barleycyc-7.0.1, accessed on 15 November 2021). However, the prediction was exclusively automatic (without a curator review) and based on the first version of the barley genome sequences. Furthermore, the mammalian melatonin biosynthesis module was used in this prediction, namely, the genes encoding the first two enzymes to convert tryptophan into serotonin were thought to be catalyzed by TPH and AADC. Two lines of clues rule against this annotation. First, both barley and rice are Gramineous family plants, and barley’s pathway could be inferred from that of rice. As concluded in two recent reviews, a set of genetic and biochemical investigations have confirmed that rice cells use TDC and T5H to generate serotonin from tryptophan [20,21]. Second, both barley and wheat are Triticeae crops, which means they are very closely related. In a recent paper using multiple detection techniques, such as the enzyme-linked immunosorbent assay (ELISA), metabolite profiling, and quantitative real-time polymerase chain reaction (qPCR), researchers proposed that wheat use TDC and T5H to synthesize serotonin [28].

In this study, we carried out a detailed in silico analysis of barley melatonin biosynthesis genes. The protein sequences of rice melatonin biosynthesis enzymes were obtained from the KEGG database. Initially, we used rice sequences as inquiries to blast against barley proteins in the NCBI database (ncbi.nlm.nih.gov, accessed on 20 December 2021) by the online software Basic Local Alignment Search Tool (BLAST). The E-values for TDC, T5H, and ASMT were set to 0, and for COMT and SNAT, they were set to less than e^−100^ and e^−87^, respectively. The identified barley sequences were then used to blast against the latest version of barley genome sequences in the EnsemblPlants (http://plants.ensembl.org/, accessed on 20 December 2021) database to yield a formal identity for the genes encoding these enzymes. We found that the barley genome has 13 *TDC*, 1 *T5H*, 1 *ASMT*, 13 *COMT*, and 2 *SNAT* genes in the melatonin biosynthesis pathway (Table 1). The number of melatonin biosynthesis genes among four major cereal crops was compared to reveal possible patterns of gene duplication or gene loss during evolution. When compared with rice, barley has a conserved number of the *T5H* and the *SNAT* genes, but a huge expansion of the *TDC* and the *COMT* gene families (Table 1). Bread wheat is an allohexaploid with the AABBDD genome, and, not surprisingly, bread wheat has three sets of melatonin biosynthesis genes, whereas the *T5H* gene family has four members because of a further gene duplication in the B genome (Table 1). Based on the protein sequences of T5H homologs in cereal crops and other plant species, a phylogenetic tree was also constructed by the Neighbor-Joining method using the MEGA11 software. As shown in Figure 2, the phylogenetic profiles agree very well with the evolutionary relationships of these organisms.

Methods to measure the melatonin content in barley and other plant species have been established [29,30,31]. As reported, barley has higher melatonin contents than Arabidopsis and rice, although its implication remains unclear. Further work to detect the expression patterns of barley melatonin biosynthesis genes and to characterize their functions are required to answer this question.

## 3. Function of Melatonin for Barley Growth and Development

Unlike the situation in Arabidopsis and rice, the genetic manipulation of barley genes, namely, to overexpress or to knock out a specific gene, has only become a practice very recently (see below). Thus, in the past, exogenous melatonin application was used as the sole method to assess the effects of melatonin on barley growth and development.

Results from early studies have inspired researchers to speculate that melatonin is an auxin-like hormone in barley. As reported in barley seedlings, melatonin displays a growth-modulating pattern similar to that of the typical plant hormone indole-3-acetic acid (IAA); that is, at higher concentrations, it promotes the elongation of etiolated coleoptiles but inhibits the growth of primary roots [32]. In addition, to implement the growth-promoting function, both melatonin and IAA induce apoplastic acidification upon pharmacological application [32]. Finally, both melatonin and IAA are derived from tryptophan, and the natural contents of melatonin in barley are approximately three-fold as abundant as IAA [32]. In a recent work using the model plant Arabidopsis as a starting material, researchers revealed two new associations between melatonin and IAA during the regulation of root growth [33]. First, as shown by transcriptome analysis, most IAA-regulated genes are co-regulated by melatonin, although the concentration of melatonin should be 1000 times higher than that of IAA. Second, in the presence of auxin biosynthesis inhibitors or auxin polar transport inhibitors, melatonin exerted minimal effects on the regulation of root growth [33]. Taken together, it seems that melatonin is not an auxin-like hormone, but it modulates root growth in an IAA-dependent manner.

The second function of melatonin in barley is to protect the photosynthesis apparatus from premature senility and functional maladjustment under multiple stress conditions. Melatonin inhibits dark-induced or abscisic acid (ABA)-induced chlorophyll degradation and leaf senescence in detached barley leaves [34]. To slow down the degradation of photosynthetic pigments during the aging of intact barley leaves, melatonin causes a reduction in the intensity of oxidative stress by maintaining a high level of pronounced antioxidants, namely, carotenoids [35]. It has been found that the drought-priming methodology (consisting of a moderate and short dehydration pretreatment) is capable of inducing an increase in endogenous melatonin production. Exogenous melatonin application was shown to induce the biosynthesis of ABA, a vital plant hormone for cold and drought resistance. Thus, the joint use of drought priming and exogenous melatonin treatment led to efficient sustainment of the photosynthetic electron transport in photosynthetic apparatus, which maintained barley biomass production under drought and cold stress conditions [36]. Melatonin pretreatment was also shown to cause a reduction in the concentration of hydrogen peroxide (H_2_O_2_) while leading to the enhancement of the activities of superoxide dismutase (SOD), ascorbate peroxidase (APX), and catalase (CAT) [36]. Heavy metals in soil lead to a decrease in the plant yield and create pollution in the food chain. Melatonin reduces the toxic effect of polymetals (several different heavy metals) on biomass accumulation by maintaining the level of carotenoids and causing an increase in the activity of SOD in barley leaves [37]. Interestingly, the effectiveness of melatonin does not depend on the duration of exposure, indicating that melatonin is a plant-priming inducer [37]. Soil salinity is one of the most global environmental constraints for crop growth. Two recent studies concerning wheat were instructive for growing barley. Combined treatments of melatonin and salicylic acid (SA) alleviate the salt-induced decrease in wheat productivity by maintaining the proper function of the photosynthetic machinery [38] and by causing an increase in the polyamine content and nitrogen metabolism [39].

The third and recently emerging function of melatonin is the regulation of the circadian rhythm in barley. In mammals, melatonin is produced according to the circadian rhythm; besides, one of the well-documented clinical applications of melatonin is modulating circadian rhythms in the treatment of sleep disorders [7,8,9]. During evolution, plants also established sophisticated circadian rhythms. As the first report in plants, the accumulation of melatonin was shown to be under a daily rhythm in *Chenopodium rubrum* [40]. Recently, it has been shown that, under cold stress, exogenous melatonin could help barley seedlings restore the oscillatory expression of circadian clock genes and then improve barley growth [41]. In rice, the expression of key melatonin biosynthetic pathway genes displays both a circadian rhythm pattern and a clear responsiveness to various abiotic stress treatments, such as drought, salt, and cold [42]. As mentioned above, the expression pattern of the barley melatonin biosynthesis genes remains unknown, but such work is urgently needed. On the other hand, it has long been recognized that the poor efficiency of the barley tissue culture system has led to a situation in which the genetic transformation and genomic editing of barley are not as routine as in other crops. Interestingly, to explain the seasonal effect on the tissue culture response and plant regeneration frequency of barley explants, researchers have speculated that the variation in the endogenous melatonin contents may be a candidate determinant that deserves to be investigated further [43].

## 4. Integrated Mechanisms of Melatonin in Plant Cells

For mammalian cells, two clear mechanisms of melatonin have long been established: (1) acting as an antioxidant, regulation of redox homeostasis, and signaling by scavenging various free radicals [7,8,25], and (2) acting as a neurohormone and the activation of transmembrane melatonin receptors [9,10,11]. In contrast, the mechanisms of melatonin in plant cells are fragmentary and have only come under systemic investigation in recent years.

### 4.1. Melatonin Serves as an Antioxidant in Plant Cells

As in animal cells, melatonin also serves as an antioxidant in plant cells [21,44]. Reactive oxygen and reactive nitrogen species (ROS and RNS, respectively) are mainly generated in chloroplasts and mitochondria under normal photosynthetic and respiration conditions. Coincidentally, melatonin is also biosynthesized in these two organelles, allowing melatonin to play a pivotal role in the regulation of free radical homeostasis inside these organelles. ROS and RNS are both signaling molecules and substances that are toxic to plant cells [16,17,18,19,20,21]. When the balance between production and scavenging is disturbed, usually under stress conditions, excessive ROS and RNS can damage cell structures and bring fatal impairment to plant cells. To perform antioxidant activity, melatonin and its metabolites make a direct connection with ROS and RNS. Through this non-enzymatic system, they exhibit high antioxidant activity, which gives one molecule of melatonin the capability of scavenging up to 10 molecules of ROS/RNS [21]. Besides, the activities of the main antioxidative enzymes, such as glutathione peroxidase and SOD, and the contents of certain antioxidants, such as ascorbic acid and glutathione, are also indirectly modulated by melatonin [44]. As a result, melatonin application could lead to a reduction in the content of malondialdehyde (MDA), a stable product of membrane lipid peroxidation, to maintain membrane integrity and to maintain cell structure [41]. Indeed, a set of studies reported that exogenous melatonin application could reduce the level of ROS and RNS in diverse plant species under various stress conditions, including drought, salinity, cold, heavy metals, and ultraviolet B (UV-B) light [45,46,47,48,49,50,51,52]. As important progress, two recent reports on Arabidopsis have revealed the physiological roles of endogenous melatonin. Molecular hydrogen (H_2_) was shown to promote melatonin biosynthesis to establish redox homeostasis and salinity tolerance in wild-type plants, but not in the *snat* mutant [52], while the overexpression of the *SNAT* gene leads to enhanced melatonin biosynthesis and redox balance under UV-B-induced stress [50].

In contrast, a set of reports proposed that melatonin could induce the production of ROS to improve the stress tolerance of plant cells. In Arabidopsis, the respiratory burst nicotinamide adenine dinucleotide phosphate (NADPH) oxidase (RBOH) was required for melatonin-induced ROS production in salt tolerance [53], stomatal closure regulation [54], and lateral root development [55]. The improvement of rice salinity stress tolerance by melatonin also required RBOH [56]. In watermelon (*Citrullus lanatus*), melatonin induced H_2_O_2_ accumulation and the Ca^2+^ wave to switch on the C-REPEAT BINDING FACTOR (CBF) pathway to confer plant cold tolerance [57]. A timely study explained the conflicting effects of melatonin on the scavenging or generation of ROS. It was determined that plant cells can efficiently convert melatonin into 2-hydroxymelatonin, and it is 2-hydroxymelatonin, rather than melatonin, that is responsible for RBOH-dependent ROS production [58].

Photosynthesis in chloroplasts is one of the most primary physiological processes for plant biomass production. Many studies have reported that melatonin can protect the photosynthetic machinery from damage under various stress conditions [34,35,36,37,38,39,59,60,61,62,63,64,65,66,67]. The function of melatonin in preventing chloroplasts from undergoing oxidative damage could mainly be related to its antioxidant features, both directly and indirectly. As mentioned above, melatonin helps barley cells maintain high levels of antioxidant carotenoids inside chloroplasts [35,37]. Under chilling stress, melatonin-deficient tomato plants generated by the virus-induced gene silencing of the *TDC* gene showed impaired antioxidant capacity and impaired photosynthesis [67]. In soybean, to maintain proper chlorophyll contents and photosynthetic rates under salt and drought stress conditions, melatonin led to the down-regulation of the expression of genes encoding chlorophyll-degrading enzymes (for example, pheophorbide A oxygenase) while leading to the up-regulation of the expression of genes encoding certain important subunits of photosystem I (*PsaK* and *PsaG*) and photosystem II (*PsbO* and *PsbP*) [62].

Polyamines are molecules that are essential for plant stress responses by the structural preservation of nucleic acids and proteins and by the modulation of the activity of a set of enzymes [68,69,70]. As with melatonin, polyamines are antioxidants capable of scavenging free radicals to maintain membrane integrity [68,69,70]. In wheat, melatonin could lead to an increase in the polyamine content (mainly three forms: putrescine, spermidine, and spermine) through the induction of their synthesis and reduction of their degradation. The expressions of genes encoding arginine decarboxylase and ornithine decarboxylase (ADC and ODC, respectively), two key enzymes for polyamine synthesis, were elevated; meanwhile, the expressions of genes encoding diamine oxidase and polyamine oxidase (DAO and PAO, respectively), which are responsible for the decomposition of polyamines, were inhibited [39]. Similar results have been reported in tomato [68], alfalfa [69], and carrot [70]. In addition, under certain abiotic stress conditions, such as low temperatures, high salinity, and excessive heavy metals, melatonin application caused an enhancement in the accumulation of soluble sugars, free proline, and other compatible protectants for the maintenance of osmotic pressure and cell integrity [39,41,66]. Figure 3 summarizes the mechanisms of melatonin in the regulation of plant abiotic stress tolerances.

### 4.2. Melatonin Modulates Plant Hormone Crosstalk

As reported in a set of plant species, as with IAA, low concentrations of melatonin promoted root elongation, while higher concentrations of melatonin inhibited root growth. However, the working concentration of IAA and melatonin could not be placed in the same category. For Arabidopsis, IAA at 0.1 to 1 nM and melatonin at 10 to 1000 nM was found to promote root growth; again, IAA at 1 nM and melatonin at 1000 nM caused changes in the expression of auxin-responsive genes in a moderately correlated manner [33]. Although the effect of melatonin seemed dependent on endogenous auxin biosynthesis and transportation, the underlying mechanism behind these effects remains unclear [33]. Tryptophan is a common precursor of melatonin and IAA biosynthesis. In certain tissues or organs, the possibility that high concentrations of exogenous melatonin inhibit the biosynthesis of endogenous melatonin exists, leading to more free tryptophan for auxin biosynthesis. Future work is required to test this speculation (Figure 3).

Recently, the reason why higher concentrations of auxin promote cell expansion in shoots but inhibit cell expansion in roots and why much lower concentrations of auxin are capable of promoting root growth was revealed. Auxin activates two antagonistically acting signaling pathways in roots: (1) a cell surface-based TRANSMEMBRANE KINASE1 (TMK1) pathway to activate plasma membrane H^+^-ATPases to induce apoplast acidification and cell expansion and (2) the intracellular auxin signaling pathway via the TRANSPORT INHIBITOR RESPONSE1 and AUXIN-SIGNALING F-BOX (TIR1/AFB) auxin receptors to induce net cellular H^+^ influx and apoplast alkalinization [71]. To explore the mechanism of melatonin on cell expansion, it is an interesting question as to whether melatonin promotes the binding of auxin with TMK1 or TIR1/AFBs.

Reports on crosstalk between melatonin and other plant hormones are not rare, but the internal mechanism remains unknown. Ethylene is especially related to the ripening of fruits, and melatonin could delay the ripening process in banana by repressing ethylene biosynthesis [72]. In tomato, it was verified that the expressions of *ACS4*, an ethylene biosynthesis gene, and *NR*, *ETR4*, *EIL1*, *EIL3*, and *ERF2*, five ethylene signaling-related genes, were enhanced after melatonin treatment [73]. Reports on the effects of melatonin on ABA biosynthesis appear to be contradictory. In barley seedlings, melatonin was shown to promote ABA biosynthesis under drought priming and cold stress conditions [36], while, in Chinese cabbage, melatonin caused the inhibition of ABA accumulation through the transcription factor ABF [74]. In apple plants, melatonin caused a reduction in the ABA content to prevent stomatal closure by causing a decrease in the expression of *MdNCED3*, an ABA synthesis gene, and an increase in the expression of *MdCYP707A1* and *MdCYP707A2*, two ABA catabolic genes [75]. Melatonin also has been reported to regulate salicylic acid (SA), jasmonic acid (JA), cytokinin, and brassinosteroid biosynthesis [76,77,78,79]. Considering that plant hormones themselves possess a complicated metabolism, signaling, and crosstalk network, it is not surprising that melatonin modulates the biosynthesis and signaling of such a large number of plant hormones (Figure 3).

Melatonin leads to the enhancement of innate immunity against infection by various pathogens, possibly through the modulation of SA and JA signaling cascades [75,76,77]. In Arabidopsis, genes related to the SA and JA signaling pathways were upregulated by exogenous melatonin [75]. Mutant plants defective in SA signaling exhibited a suppressed effectiveness of melatonin application [75,76]. Through the activation of the SA signaling pathway, melatonin induced the expression of marker genes encoding pathogenesis-related (PR) proteins [77]. Figure 4A illustrates this mechanism of melatonin on the regulation of biotic stress tolerances.

### 4.3. Putative Plant Melatonin Receptor CAND2/PMTR1

Human cells possess two G-protein-coupled receptors (GPCRs) that bind melatonin [9,10,11]. Whether plants have melatonin-binding GPCRs is a very interesting question. The heterotrimeric G protein complex of mammalian cells contains one subunit of each of Gα, Gβ, and Gγ, and when this complex is activated by a GPCR, GTP is exchanged for GDP on Gα. Plant cells have the three subunits of the G protein complex and some putative GPCRs. Since plant GPCRs do not possess guanine nucleotide exchange factor activities, the G protein signaling mechanism of plant cells is quite different from that of animal cells [80]. In a pioneering study, two plant GPCR candidates, CAND2 and CAND7, were reported to be involved in Arabidopsis root growth regulation by perceiving *N*-acyl-homoserine lactones, which were secreted by the rhizosphere bacteria [81]. In 2018, CAND2 was suggested as a phyto-melatonin receptor1 (PMTR1) by several lines of evidence. As strong evidence, the *cand2* mutant of Arabidopsis was insensitive to melatonin-induced stomatal closure [82]. However, another research group proposed that CAND2 was not a bona fide GPCR-type melatonin receptor, the CAND2 protein was localized to the cytoplasm rather than the plasma membrane, and the *cand2* mutant was competent in the melatonin-mediated mitogen-activated protein kinase (MAPK) signaling pathway to establish the defense responses of Arabidopsis plants to pathogens [83].

Nevertheless, two recent reports provided additional evidence for the role of CAND2/PMTR1 in perceiving melatonin. CAND2/PMTR1 is a plasma membrane protein capable of transducing signals from the FLS2/BAK1 (FLAGELLIN SENSING2/BRASSINOSTEROID INSENSITIVE1-ASSOCIATED KINASE1) complex to prime stomatal immunity against bacterial invasion [84]. The MAPK signaling cascade and the Gα subunit GPA1 signaling pathway are located downstream of the CAND2/PMTR1 protein. They promote the expression of *MPK3/6* and the phosphorylation of MPK3/6; meanwhile, they also directly interacted with GPA1 to support its ability in the activation of RBOH [84]. In a separate report, researchers proposed that the function of melatonin is to regulate redox homeostasis, and abiotic stress tolerance may also be related to the CAND2/PMTR1 protein, considering that the *cand2* mutant of Arabidopsis exhibited a reduction in osmotic stress tolerance and increased ROS accumulation [85]. Figure 4B illustrates the role of CAND2/PMTR1 in the regulation of stomatal immunity by melatonin.

## 5. Application of Melatonin in Barley

### 5.1. Melatonin for Barley Biotic and Abiotic Stress Tolerance

As discussed above, melatonin is a pleiotropic molecule that protects plants against various stresses. A large amount of barley is cultivated in adverse circumstances, including drought and cold climates, which produces a low yield. In addition, barley is often subject to harmful diseases, such as barley yellow mosaic virus, barley powdery mildew, and barley net blotch disease. The application of exogenous melatonin in barley cultivation should enhance stress tolerance and expand the geographical limits of cultivability. Currently, treatments of barley plants grown in a greenhouse with melatonin solutions by foliage spraying or root watering have been reported [36,37]; however, such studies have lagged behind when compared with other crops, and the treatment duration and the effective dosage of exogenous melatonin for barley need more solid investigation.

An alternative and promising approach is to modulate the melatonin levels in barley using genetic and genomic engineering. Recently, stable and reliable techniques for the genetic manipulation of barley were established. By transcriptional profiling, the key regulatory factors related to the formation of embryogenic callus in barley were identified [86]. In addition, using the ectopic expression of certain transcription factor genes, the plant regeneration capacity was improved to three times [86]. The overexpression of barley *Ant1*, a transcription factor gene controlling anthocyanin biosynthesis, led to the enrichment of anthocyanin accumulation in transgenic barley grains [87]. Furthermore, a clustered, regularly interspaced short palindromic repeats (CRISPR)/Cas9 genome editing system was developed for barley to modulate the content of the antioxidant vitamin E in barley grains [88]. In the future, these systems will be used to knock out or overexpress melatonin biosynthesis genes in barley and to obtain a clearer understanding of the function and mechanism of melatonin in barley growth and development.

As previously reported, the overexpression of the *SNAT* gene (encoding a rate-limiting enzyme in the mammalian melatonin biosynthetic pathway) of sheep in rice by the maize *Ubiquitin-1* promoter led to 10-fold higher levels of melatonin accumulation; meanwhile, the transgenic rice plants had stronger roots and earlier seedling growth, but delayed flowering and severely reduced grain yield [89]. The hyperactive synthesis of melatonin may impact other metabolic pathways, for example, auxin biosynthesis, whose starting substrate is also tryptophan. It has been shown that, to improve plant stress tolerance, the use of strong constitutive promoters, such as *35S* and *Ubi-1*, usually results in severe side effects under normal growth conditions. It was found that the use of the stress-inducible *RD29A* promoter could eliminate melatonin-induced side effects [90]. Thus, to improve the melatonin levels in barley, stress-inducible promoters or endosperm-specific promoters should be used according to the aims of the studies.

### 5.2. Melatonin in Beer and Food

Melatonin has multiple healthcare functions for mammals and is widely used to relieve certain diseases on immunity, circadian rhythms, and sleep [9,10,11]. Recently, its beneficial effects in the treatment of various cancers, particularly in the case of chemical and radiological therapies, were also shown [16]. The safety of exogenous melatonin in humans has been evaluated in depth [91]. Short-term use of melatonin is safe even in extreme doses, and long-term use causes only mild adverse effects [91]. However, currently, commercially used melatonin is chemically synthesized. With rapid advances in plant melatonin research and the demand for chemical-free foods, a research objective addressing how plant melatonin benefits human health is beginning to emerge. Barley is a unique cereal crop popularly used for beer, food, and feed [23,24], and the topic of how barley melatonin relates to human health is discussed below.

It has been reported that melatonin in beer contributes to the increased melatonin level and antioxidant capacity of human serum [92]. Under growth and fermentation conditions, various yeast strains could synthesize melatonin [93]. Thus, high levels of melatonin are generated by barley and yeast in the malting and fermentation processes, respectively [94]. During bread dough fermentation, the formation of melatonin has also been observed [95]. Melatonin from dietary intake has protective effects against cardiovascular and neurodegenerative diseases [96]. A recent review provided a detailed introduction of the bioavailability, pharmacokinetics, and beneficial effects of melatonin from beer and wine [97]. 

## 6. Conclusions and Perspectives

Melatonin regulates various aspects of plant physiological processes, including seed germination, shoot and root growth, flowering time and circadian rhythm, fruit ripening and flavoring, and others. It improves plant growth and development under multiple abiotic or biotic stress conditions. It also benefits the food industry. For example, as a rapidly developing field, melatonin was used in the postharvest storage of fruits and vegetables to extend their shelf life by delaying the ripening process, alleviating chilling injury, and inhibiting the decay process [44,98,99]. Three mechanisms found in plant melatonin are discussed: (1) serving as an antioxidant, (2) modulating plant hormone crosstalk, and (3) signaling through the putative plant melatonin receptor CAND2/PMTR1. To achieve the beneficial effects of melatonin on plant cells, these mechanisms should work together.

The function and mechanism of melatonin in barley remain largely unknown. It is worth learning from advanced studies on Arabidopsis and rice that consider barley as a unique crop for food, feed, and malt. In the present paper, we carried out in silico analysis to identify barley melatonin biosynthesis genes. This analysis should be helpful for expanding the investigation into barley melatonin. Several important research areas or topics are raised for reference: (1) whether the expression of barley melatonin biosynthetic pathway genes display a circadian rhythm or seasonal pattern, and whether the tissue culture response and plant regeneration frequency of barley explants are related to the melatonin contents; (2) how to reveal the physiological and pathological functions of barley melatonin biosynthesis genes by genetic engineering and how to improve the melatonin contents in barley grains and in beer products without side-effects; and (3) do barley CAND2/PMTR1 homologs serve as melatonin receptors? Does melatonin modulate auxin biosynthesis and transportation in barley? The increasing reliability of genomic editing techniques should help to answer these questions.

## Figures and Tables

**Figure 1 antioxidants-11-00634-f001:**
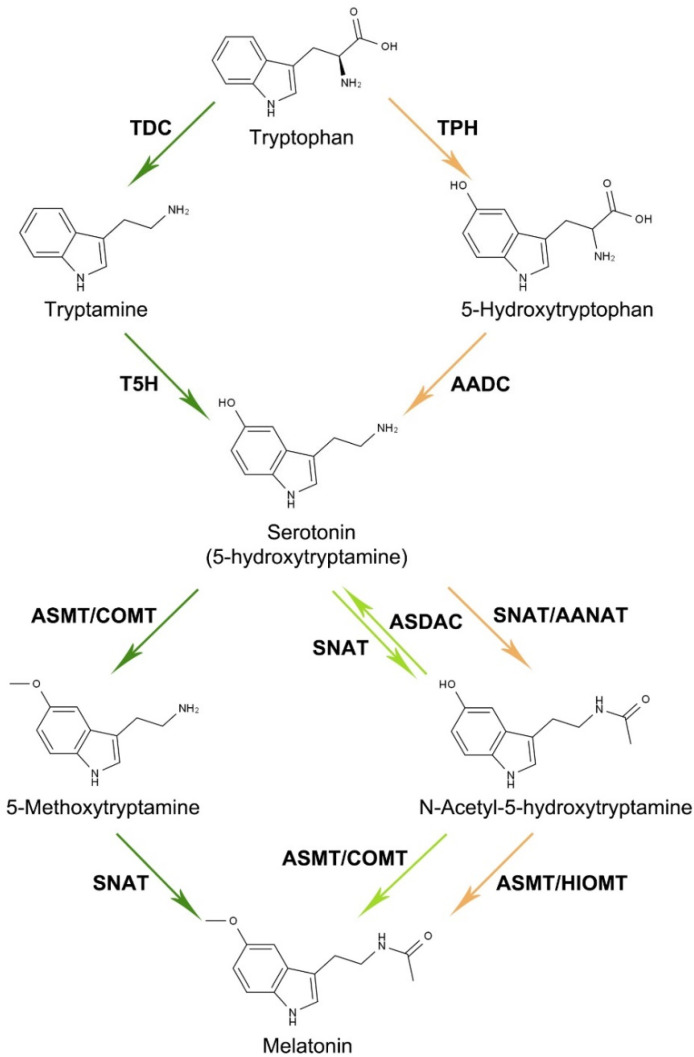
Comparison of the melatonin biosynthesis pathways in animals and plants. The yellow arrows indicate the pathway of animals, while the green arrows indicate the pathway of plants. Plant SNATs are rate-limiting enzymes catalyzing two reactions. The light green arrows indicate a minor pathway, since the SNAT enzymes of most plants tend to accommodate 5-Methoxytryptamine as the prevailing substrate, rather than serotonin [21,27]. TDC, tryptophan decarboxylase; TPH, tryptophan hydroxylase; T5H, tryptamine 5-hydroxylase; AADC, aromatic amino acid decarboxylase; COMT, caffeic acid *O*-methyltransferase; SNAT, serotonin *N*-acetyltransferase; ASDAC, *N*-acetylserotonin deacetylase; AANAT, arylalkylamine *N*-acetyltransferase; ASMT, *N*-acetyl serotonin methyltransferase; HIOMT, hydroxylindole-*O*-methyltransferase.

**Figure 2 antioxidants-11-00634-f002:**
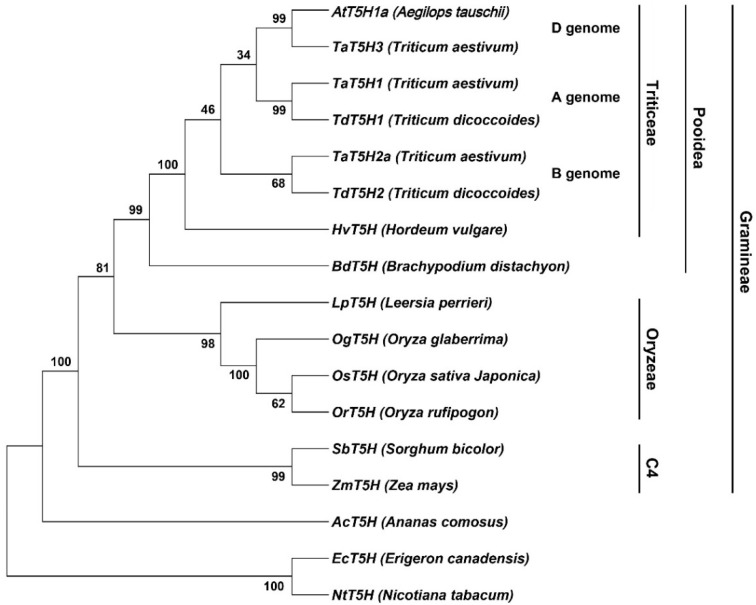
A phylogenetic tree of tryptamine 5-hydroxylase (T5H) homologs in cereal crops and other plant species. The picture was constructed by the Neighbor-Joining method using the MEGA11 software. Bread wheat is an allohexaploid with the AABBDD genome, and, not surprisingly, bread wheat has three sets of melatonin biosynthesis genes. The phylogenetic profiles agree very well with the evolutionary relationships of these organisms.

**Figure 3 antioxidants-11-00634-f003:**
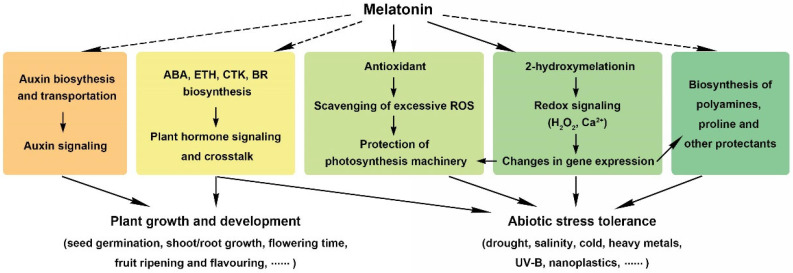
The function and mechanism of melatonin in the regulation of plant growth, development, and abiotic stress tolerance. To regulate seed germination, shoot/root growth, flowering time, and fruit ripening and flavoring, melatonin mainly modulates plant hormone crosstalk. To regulate drought, salinity, cold, heavy metals, UV-B, and nanoplastics tolerance, melatonin serves as an antioxidant in plant cells and induces the biosynthesis of polyamines, proline, and other protectants. The dotted lines indicate that the mechanism is unclear.

**Figure 4 antioxidants-11-00634-f004:**
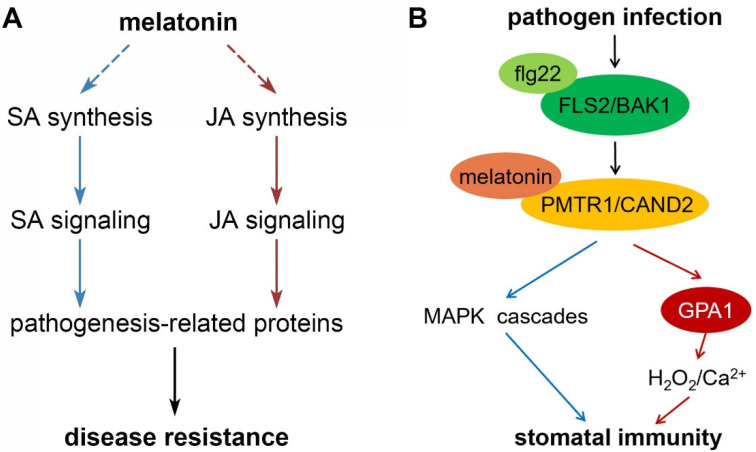
Two mechanisms of melatonin on the regulation of plant disease resistance. (**A**) Melatonin modulates the salicylic acid (SA) and jasmonic acid (JA) signaling cascades to enhance plant innate immunity against infection by various pathogens. The dotted lines indicate that the detailed processes remain unknown. (**B**) The PMTR1/CAND2 pathway regulates stomatal closure under pathogen infection. flg22: highly conserved N-terminal of flagellin epitope, recognized by FLS2; FLS2/BAK1: Flagellin Sensing 2/Brassinosteroid Insensitive 1-associated kinase 1; GPA1: G protein α subunit; PMTR1/CAND2: phyto-melatonin receptor; MAPK: mitogen-activated protein kinase.

**Table 1 antioxidants-11-00634-t001:** Number of genes for the melatonin biosynthesis pathway in four major cereal crops.

	Rice *Oryza sativa*	Barley *Hordeum vulgare*	Wheat *Triticum aestivum*	Maize *Zea mays*
*TDC*	4	13	~39	5
*T5H*	1	1	4	2
*ASMT*	3	1	3	0
*COMT*	1	13	~39	1
*SNAT*	2	2	6	2

TDC, tryptophan decarboxylase; T5H, tryptamine 5-hydroxylase; ASMT, *N*-acetyl serotonin methyltransferase; COMT, caffeic acid *O*-methyltransferase; SNAT, serotonin *N*-acetyltransferase. The formal identities for the barley genes encoding these enzymes are listed below. For clarity, the prefix “HORVU.Morex.r3” (for species *Hordeum vulgare* L, cultivar Morex, the third version reference sequence) before the barley chromosome number (1H, 2H, 3H, 4H, 5H, 6H, and 7H) is omitted. *HvTDC1*, 2HG0205310; *HvTDC2*, 5HG0459390; *HvTDC3a*, 7HG0643110; *HvTDC3b*, 7HG0642940; *HvTDC3c*, 2HG0205290; *HvTDC3d*, 1HG0010600; *HvTDC3e*, 2HG0205260; *HvTDC3f*, 7HG0643120; *HvTDC3g*, 7HG0721890; *HvTDC3h*, 3HG0224270; *HvTDC4a*, 3HG0305100; *HvTDC4b*, 3HG0305050; *HvTDC4c*, 3HG0305070. *HvT5H*, 6HG0554760. *HvASMT*, 1HG0002020. *HvCOMT1*, 7HG0711890; *HvCOMT2*, 6HG0540090; *HvCOMT3*, 3HG0330120; *HvCOMT4*, 1HG0089900; *HvCOMT5*, 4HG0332150; *HvCOMT6*, 1HG0003780; *HvCOMT7*, 2HG0210170; *HvCOMT8a*, 1HG0012390; *HvCOMT8b*, 1HG0012400; *HvCOMT8c*, 1HG0012410; *HvCOMT8d*, 1HG0012420; *HvCOMT8e*, 1HG0012430; *HvCOMT9*, 7HG0749230. *HvSNAT1,* 1HG0070690; *HvSNAT2,* 7HG0707850.

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
