# Peer review of "Function, Mechanism, and Application of Plant Melatonin: An Update with a Focus on the Cereal Crop, Barley (Hordeum vulgare L.)"

_antioxidants, 2022, doi:10.3390/antiox11040634_

Round 1

Reviewer 1 Report

Comments:

The topic –plant melatonin- is an interesting, modern, new and evolving field of plant biology. Our knowledge about this multifunctional hormone-like molecule in the economically important barley has been very limited. The authors' summary of the results so far, supplemented by their own findings, is useful and forward-looking. The authors are sufficiently critical of the relevant literature, and their critical remarks are well-founded.

Some critical remarks:

Line 169: Please, change: Hordeum Vulgare L.   to Hordeum vulgare L.

Line 191: Instead of “have become practical” in my opinion it might be more expressive:  “have become practice”

Line 295: Please correct.” convert melatonin into 2-hydroxymelationin; and it is 2-hydroxymelationin…..”   to: „convert melatonin into 2-hydroxymelatonin; and it is 2-hydroxymelatonin” Please correct the text in Figure 3. as well!

Lines 403-404: Please, instead of „located downstream” term, explain more precisely the relationship of melatonin-PMTR1 to the MAPK- and GPA1-cascades.

Line 424: Instead of “planting score” I suggest: "geographical limits of cultivability” OR “area of cultivability”

Lines 430-432: This sentence means, that the ability to regenerate has tripled? = improved to three times?

My question: Are there any in vivo, even outdoor experiments with exogenous melatonin treatments of barley at all?

Author Response

Dear reviewer,

Thank you very much for reviewing our manuscript entitled "Function, mechanism, and application of plant melatonin: an update with a focus on the cereal crop, barley (Hordeum vulgare L.) " for Antioxidants (Manuscript ID# 1648548). We have revised our manuscript based on your suggestions. A 'Track Changes' version of the revision is uploaded to clearly indicate the changes.

The changes we have made is outlined below.

1, The spelling or verbal errors in Line 169, Line 191, Line 295, Lines 403-404, and Lines 430-432 have been corrected.

2, Lines 403-404: Please, instead of “located downstream” term, explain more precisely the relationship of melatonin-PMTR1 to the MAPK- and GPA1-cascades.

A short description has been added as follows:

It promoted the expression of MPK3/6 and the phosphorylation of MPK3/6, at meanwhile, it also directly interacted with GPA1 to support its ability on activation of RBOH [84].

3, My question: Are there any in vivo, even outdoor experiments with exogenous melatonin treatments of barley at all?

We have rewritten the sentence as follows:

Currently, treatments of barley plants grown in greenhouse with melatonin solutions by foliage spraying or root watering have been reported [36, 37], however,

We hope the revised manuscript is suitable for publication in Antioxidants.

Thank you very much for your consideration.

Sincerely yours,

Junhui Wang

Reviewer 2 Report

To the Authors,

The aim of this review (Manuscript ID: antioxidants-1648548) is to focus the mechanisms in which plant melatonin are involved, pointing out melatonin biosynthesis and functions in barley. The topic of the review is of considerable interest for the scientific community. The review include comprehensive and critical view of the research area, however some points are not so clear in the melatoin biosynthetic pathway in plant cells.

I suggest to clarify:

Page 3 Lines 106-107: please add some references or clarify the role of ASDAC and ASMT/HIOMT in plants. Is N-Acetyl-5-hydroxytryptamine an intermediate of the biosynthesis of melatonin also in plants? After, in Figure 1 are reported light green arrows, what are they meaning?

Page 4 Line 120-121: It is better define chloroplasts as organelles, they are not properly sites.

Author Response

Dear reviewer,

Thank you very much for reviewing our manuscript entitled "Function, mechanism, and application of plant melatonin: an update with a focus on the cereal crop, barley (Hordeum vulgare L.) " for Antioxidants (Manuscript ID# 1648548). We have revised our manuscript based on your suggestions. A 'Track Changes' version of the revision is uploaded to clearly indicate the changes.

The changes we have made is outlined below.

1, Page 3 Lines 106-107: please add some references or clarify the role of ASDAC and ASMT/HIOMT in plants. Is N-Acetyl-5-hydroxytryptamine an intermediate of the biosynthesis of melatonin also in plants? After, in Figure 1 are reported light green arrows, what are they meaning?

We have added some reference citations in the text and some new description to the legend of Figure 1 as follows:

Plant SNATs are rate-limiting enzymes catalyzing two reactions. The light green arrows indicate a minor pathway, since SNAT enzymes of most plants tend to accommodate 5-Methoxytryptamine as the prevailing substrate than serotonin [21, 27].

2, Page 4 Line 120-121: It is better define chloroplasts as organelles, they are not properly sites.

We have corrected it.

We hope the revised manuscript is suitable for publication in Antioxidants.

Thank you very much for your consideration.

Sincerely yours,

Junhui Wang
